# Interplay of Gut Microbiota in Polycystic Ovarian Syndrome: Role of Gut Microbiota, Mechanistic Pathways and Potential Treatment Strategies

**DOI:** 10.3390/ph16020197

**Published:** 2023-01-28

**Authors:** Leander Corrie, Ankit Awasthi, Jaskiran Kaur, Sukriti Vishwas, Monica Gulati, Indu Pal Kaur, Gaurav Gupta, Nagavendra Kommineni, Kamal Dua, Sachin Kumar Singh

**Affiliations:** 1School of Pharmaceutical Sciences, Lovely Professional University, Phagwara 144411, India; 2ARCCIM, Faculty of Health, University of Technology Sydney, Ultimo, NSW 2007, Australia; 3University Institute of Pharmaceutical Sciences, Panjab University, Chandigarh 160014, India; 4School of Pharmacy, Suresh Gyan Vihar University, Mahal Road, Jaipur 302017, India; 5Department of Pharmacology, Saveetha Dental College, Saveetha Institute of Medical and Technical Sciences, Saveetha University, Chennai 600007, India; 6Uttaranchal Institute of Pharmaceutical Sciences, Uttaranchal University, Dehradun 248007, India; 7Center for Biomedical Research, Population Council, New York, NY 10065, USA; 8Discipline of Pharmacy, Graduate School of Health, University of Technology Sydney, Ultimo, NSW 2007, Australia

**Keywords:** PCOS, gut microbiota, prebiotic, probiotic, polyphenol

## Abstract

Polycystic Ovarian Syndrome (PCOS) comprises a set of symptoms that pose significant risk factors for various diseases, including type 2 diabetes, cardiovascular disease, and cancer. Effective and safe methods to treat all the pathological symptoms of PCOS are not available. The gut microbiota has been shown to play an essential role in PCOS incidence and progression. Many dietary plants, prebiotics, and probiotics have been reported to ameliorate PCOS. Gut microbiota shows its effects in PCOS via a number of mechanistic pathways including maintenance of homeostasis, regulation of lipid and blood glucose levels. The effect of gut microbiota on PCOS has been widely reported in animal models but there are only a few reports of human studies. Increasing the diversity of gut microbiota, and up-regulating PCOS ameliorating gut microbiota are some of the ways through which prebiotics, probiotics, and polyphenols work. We present a comprehensive review on polyphenols from natural origin, probiotics, and fecal microbiota therapy that may be used to treat PCOS by modifying the gut microbiota.

## 1. Introduction

Polycystic ovarian syndrome (PCOS) is a complex metabolic disorder. It combines various symptoms such as anovulation, hirsutism, amenorrhea, infertility, obesity, and polycystic ovaries. Globally, it is known to affect 6% to 20% of women [1]. PCOS is associated with various other diseases, including obesity, diabetes, cardiovascular implications and cancers. This has led to an increased economic burden and has attracted interest in this field [2]. The therapeutic options available for PCOS include a change in lifestyle, diet, exercise, and pharmacotherapy. However, the standard pharmacological approaches have not given satisfactory results, and PCOS prevalence is still on the rise [3]. In recent years, the strong association of gut microbiota with physiology of female reproductive functions has been reported [4]. Research suggests that there is a link between gut microbiota and the host metabolism, through various metabolites secreted by them like short-chain fatty acid [5], timethylamine N-oxide (TMAO), and inosine-5-monophosphate (IMP), etc. These metabiotics, in turn, show their effects on supplying energy for colonic epithelial cells, formation of white adipose tissue and metabolism of lipids in the host [6]. Moreover, the gut microbiota is also known to regulate the host immune system and control the secretion of bile acids related to the digestion and metabolism of the host [7]. Dysbiosis of the gut is widely reported to be responsible for certain intestinal diseases, including; Crohn’s disease, ulcerative colitis, and inflammatory bowel disorder, that are reported to be more frequent in PCOS patients [8]. 

The association between PCOS and gut microbiota is majorly attributed to the release of endotoxins and gut inflammation. Enteroendocrine cells [less than 1% of all gastrointestinal (GI) epithelial cells] release gut hormones that play important role in the hormonal networks throughout interdigestive and postprandial periods. Over 30 such gut hormones have now been identified. This suggests that eating behavior and GI motility are collaboratively regulated by gut hormone production [9,10]. It is also reported that gram-negative bacteria cause inflammation in the gut due to the production of lipopolysaccharide [11]. Microbiome diversity is a significant facor indicative of the health of the host. Obese-PCOS individuals have lesser alpha (diversity of the microbiome relevant to a single sample) and beta diversity (diversity of the microbiota relevant to diverse samples) of microbiota as compared to lean individuals [12]. Therefore, modulation of the gut microbiota could be effective in the treatment of PCOS. Even the underlying mechanism of the diseases such as diabetes, hyperlipidemia and obesity are considered to be associated with the composition and diversity of gut microbiota. Therefore, the role of gut microbiota in other diseases having metabolic significance and relevance with PCOS also assumes very high significance [13]. 

Literature search was carried out by screening the manuscripts from the following databases; Web of science, PubMed, Scopus, Embase and Science Direct. The following combinations of keywords were used to search the literature: “Gut Microbiota”, “PCOS”, “Bacteria”, “Fungi”, “Virus”, “Ovarian”, “Ovaries”, “Obese”, “Insulin Resistance”. The manuscripts which highlighted the role of gut microbiota in PCOS as well as PCOS related metabolic abnormalities were selected and classified into the following subsections for this review. 

## 2. Gut Microbiota and the Endocrine System 

The human gut hosts numerous microbes, including bacteria, viruses, fungi, and archaea. Alteration in the composition of these species is believed to be related to change in the endocrine function of the host [14]. The microbiota in the gut is reported to play a significant role in the endocrinal physiology of humans [15]. There are many reports on the interplay of the intestinal bacteria and the sex hormones of an individual. A study by Mueller et al. in 2006, indicated an increase in *Bacteroides*, *Eubacterium*, and *Blautia* in males as well as an increase in *Treponema* in females [16]. 

There are some reports that indicate that specific gut microbes play a significant role in the synthesis of sex hormones [17,18]. Table 1 highlights the recent studies on gut microbiota and its effects on hormonal signaling. Most of these studies use the 16S ribosomal ribonucleic acid (16S rRNA) technique and metagenomic analysis for their understanding of the gut microbiota. Moreover, there are several reports on the association of gut microbiota and estrogen effects on hormone induced women reproductive health. For instance, in a cross sectional analysis in women, changes in gut microbiome were observed between postmenopausal and premenopausal women, matching the variations between men and premenopausal women. It was, therefore, inferred, that female hormone deficiency causes gut microbial alterations during menopause [19]. In another study it was found that premenopausal women had an abundance of the microbial steroid downregulation, which was positively linked with plasma progesterone levels [20]. Total urine estrogen was reported to be strongly linked with gut microbial diversity in 7 postmenopausal women and 25 men [21]. Similarly, higher gut microbial diversity was linked to serum estradiol in 26 women, but this study did not compensate for menstrual timing [22]. In another study, among 16 women, the use of combination hormonal contraceptives, which reduce serum estradiol and progesterone, was linked to a reduction in gut microbial diversity [23]. These findings imply that menopause and low estrogen are linked to a reduction in gut microbiota diversity, highlighting the close relationship of the gut microbiota community and its action on production of estrogen. 

## 3. Role of Gut Microbiota in PCOS

Steroidal hormonal levels are reported to have a relationship with the gut microbiome changes, and women suffering from PCOS exhibit higher amounts of androgens linked with a change in metabolic activity [35]. Several studies have been carried out to correlate gut microbiota, and PCOS with an overall inference that any decrease in the alpha and beta diversity of the microbiome in the gut is correlated with the occurrence of PCOS [36]. There is a link between lower alpha diversity of gut microbiota and obesity, which is one of the most prominent co-morbidies of PCOS in women [37]. Since the last decade, much research has been carried out to understand gut bacteriome’s association and its relationship with PCOS. Though consideration of the concept of the existence of a mycobiome or virome would reflect on the role of microbiome as a whole, such reports are quite limited [38,39].

### 3.1. Bacteria Involved in PCOS

#### 3.1.1. *Firmicutes*


This phylum of bacteria represents the maximum diversity within the human gut [40]. *Firmicutes* mainly comprise *Lactobacillus*, *Clostridium*, and *Ruminococcus*. Of all the phyla of *Firmicutes*, the effect of *Lactobacillus* on human health has been extensively studied, and its direct relationship with PCOS has been established [41]. The ratio of Firmicutes and Bacteroidetes has been correlated to gut microbiota in obese people [42]. A study by Liu et al. found a decrease in *Ruminococcaceae* and *Clostridium* in the PCOS group who were obese [43]. Another study found a decrease in *Clostridiales* [44] while another study by Insenser et al. confirmed a higher abundance of *Lachnospiraceae oribacterium* [45]. 

The genus *Lactobacillus* consists of many species, some of which are related to obesity and PCOS. These bacteria have been associated with a lack of dextrin synthases [46]. A close relationship has been reported between weight gain and the abundance of *L. acidophilus* species in an individual [47]. In a study involving Sprague dawley rats, *L. plantarum* was found to increase the synthesis of isobutyric acid and isovaleric acid, which are known to play a role in lipid metabolism and are critical for PCOS [48]. Contrastingly, another study found that treatment of postmenopausal women with *L. plantarum* reduced the glucose levels and lowered C-reactive proteins in white adipose tissue [49]. The administration of *L. johnsonii* to rats increased their granulosa layers and the formation of corpora lutea alleviating PCOS [50]. In a study carried out on Iranian women. oral administration of *Lactobacillus* strains (*L. acidophilus*, *L. plantarum*, *L. fermentum*, and *L. gasseri*) for 12 weeks was found to reduce the Interleukin-6 (IL-6) and high-sensitivity C reactive protein (hs-CRP) levels, improving inflammation associated with PCOS [51]. In another study on PCOS-induced rats, *L. reuteri* was reported to improve reproductive function and restore the gut microbiota [52].

*L. reuteri* is reported to improve insulin resistance (IR) and browning of white adipose tissue [53]. In another study, reduction in blood glucose levels, an improvement in insulin sensitivity, and concentration was seen in the postpartum period in women when they were treated with *L. rhamnosus* [54]. The role of *L. acidophilus* and *L. casei* in reducing plasma glucose, improving insulin levels and increased insulin sensitivity were highlighted similarly in another report [55]. Mice that were treated with a high-fat diet (HFD) showed a decrease in body weight and downregulation of Tumor Necrosis Factor α (TNF-α) and Interleukin -1β after treatment with *L. sakei* [56] Female C57BL/6 mice, when fed with *Lactobacillus* JBD301, increased the fecal excretion and the gut fluid, inhibiting the weight gain, which is commonly seen in PCOS women [57]. Another study pointed out the role of *L. gasseri* BNR17 in decreasing visceral fat mass and waist circumference [58]. Often *Lactobacillus* is used in combination with certain prebiotics to make synbiotics for the treatment of PCOS symptoms. Some of the studies using synbiotics for treatment of PCOS are highlighted in Table 2. 

#### 3.1.2. Actinobacteria

*Actinobacteria* constitute one of those significant phyla of gut bacteria that play a significant role in its eubiosis [64]. These are primarily gram-positive, non-spore-forming, anaerobic bacteria. This phylum includes three prominent families i.e., *Bifidobacteria*, *Propionibacteria*, and *Corynebacteria* [65]. The family that is abundantly present in the human gut is *Bifidobacteria* [66]. The metabolic pathway functional in the bacteria in this phylum is the one involving fermentation of sugars, releasing hydrogen and short-chain fatty acids (SCFAs) [67]. This phylum is also involved in regulating lipid metabolism [68]. All these aspects point towards its role in the regulation of PCOS symptoms. 

Administration of *Bifidobacterium lactis* 420 to C57BL/6J mice for 12 weeks [69] resulted in weight loss. Increase in gut permeability is linked to a decreased proportion of *Bifidobacteria* [66]. This, in turn, leads to the transport of lipopolysaccharides into the blood serum, thereby triggering the immune system resulting in chronic inflammatory responses. Mice on a high-fat diet that were administered with *B. pseudocatenulatum* CECT 7765 showed a reduced production of cytokines such as IL-6 [70]. Glucose tolerance is reported to be directly linked with these specific inflammation markers. This study also highlighted the fact that after administration of the specific bacterial species, the inflammation marker IL-10 rose exceptionally high. Phagocytic function in the peritoneal macrophages was a result of oxidative stress, causing inflammation. It is pertinent to add here that the number of *Actionbacteria* and regulation of the inflammation levels are correlated to each other. Certain subspecies of *B. longum* (*B. infantis*) have shown their effect on the T cells activity in various animal models [71]. In a study involving the administration of *B. infantis* for the treatment of ulcerative colitis and psoriasis, it was found to lead to a reduction in the C reactive protein (CRP) levels [72]. Moreover, in an in vitro study using Caco-2 cells, *B. adolescentis* decreased tumor necrosis factors (TNF-α) expressions [73].

Administration of *B. pseudocatenulatum* CECT 7765 to mice showed an improvement in regulation of fatty acids and cholesterol, metabolism [74]. Treatment of mice with *B. longum* decreased the weight of mice and reduced the levels of cholesterol, triglycerides, serum aspartate transaminases, and alanine transaminases [75]. Another study involving the administration of *Bifidobacterium* MKK4 to obese mice reported reduced triglyceride and cholesterol levels and regulated the gut microbiota [76]. In a randomized control trial (RCT), administration of *B. breve* B-3 showed alterations in the levels of γ-glutamyltranspeptidase and high-sensitivity CRP that is linked with low-fat mass [77]. Another study, evaluating the role of *Bifidobacterium* levels in women, found it to be in lower abundance in the PCOS group than in the control group [78]. All these studies highlight the fact that the members of phylum, *Actinobacteria* is actively involved in various metabolic and inflammatory pathways leading to PCOS.

#### 3.1.3. Proteobacteria 

*Proteobacteria* is one of the major phyla of bacteria that are gram-negative [79]. They represent most of the pathogenic bacteria that include *Salmonella*, *Vibrio*, *Heliobacteria*, and *Yersinia*. Alink has been established between *Proteobacteria* and the imbalance caused in the lower reproductive tract of women and inflammation [80]. *Escherichia fergusonii* from this phylum is reported to be responsible for acute cystitis (an underlying risk factor for PCOS) [81]. Interestingly, some clinical data shows a relationship between acute cystitis and PCOS [82]. The genus *Salmonella* belonging to this bacterial phylum though commonly known to be associated with typhoid, has also been reported to be associated with PCOS. Certain species of *Salmonella* are known to cause egg contamination and reproductive tract infections in chickens [83].

*Salmonella typhimurium* is known to reduce ovarian cancer in metastatic and dissemination mouse models and warrants further study on its effects on PCOS [84]. *Helicobacter* bacteria belonging to this phylum have also been implicated in PCOS. *H. pylori* is known to cause inflammation in the gastric tract. However, its role PCOS was reported where the seropositivity of *H. pylori* was found higher in the PCOS group along with a higher concentration of C reactive protein than that in the control group [85].

Another pathogen found abundantly in PCOS patients is *Comamonas kerstersii*, which has been linked to peritonitis and urinary tract infections [86,87]. A recent study by Chu et al., indicated that the PCOS group had a higher abundance of the genus *Shigella* [88]. These are thought to cause disease by secreting virulence factors, producing severe inflammation, and mediating colon enterotoxic effects. They inject virulence effectors into epithelial cells to facilitate invasion of the cells and downgrade inflammation [89]. In women with PCOS, there is an increase in gram-negative bacteria such as *Desulfovibrio* [43]. higher abundance of *Desulfovibrio* bacteria is correlated to short-chain fatty acids, an increase in follicle-stimulating hormone, and IL-10 [90]. 

#### 3.1.4. Bacteroidetes

*Bacteroidetes* constitute the largest phylum of gram-negative bacteria found in the gastrointestinal system, and are considered as the key players in the gut microbiota’s healthy state and homeostasis [91]. *Bacteroides*, *Prevotella*, and *Porphyromonas* represent the three main genera of this phylum. The host’s metabolism is regulated by *Bacteroidetes*, which harvest polysaccharides and produce short-chain fatty acids [92]. In a cohort study, it was pointed out that the presence of family S24-7of *Bacteroidetes* was significantly lower in stools of PCOS patients and was associated with reproductive hormones such as thyroid stimulating hormone (TSH) and luteinizing hormone (LH) [93]. Another study indicated that *B. fragilis* was significantly higher in the PCOS group [88]. Liu and coworkers revealed that bacteria belonging to *Bacteroides* were significantly higher in the PCOS group and were negatively correlated to ghrelin and positively correlated to testosterone as well as body mass index (BMI) levels [43].

Similarly, Arroro et al., pointed out that the abundance of *Parabacteriodes* was significantly decreased when the mice were treated with letrozole, an agent used to induce PCOS [94]. In an interesting study that analyzed the gut microbial composition in 100 women from rural Ghana and urban America (50 lean women and 50 obese women), it was found that lean American women had more *Bacteroides* whereas lean Ghanaians had more butyrate-producing gut bacteria. Mice transplanted with the faeces of a lean Ghanaian were found to be resistant to obesity, produced by a 6-week high-fat diet (*p* < 0.01), highlighting the fact that this phylum can play a role in the management of weight in PCOS women [95]. The genus *Alistipes* belonging to this phylum and family *Rikenellaceae* was reported to be associated with increased gut inflammation and showed a negative correlation with LDL and a positive correlation with triglycerides in rats induced with PCOS on an ordinary diet [96]. 

### 3.2. Fungi Involved in PCOS

Characterizing the human gut mycobiome has been aided by next-generation sequencing techniques [97,98]. More than 66 genera and 184 species of fungi have been discovered in the human gut, with *Candida*, *Saccharomyces*, and *Cladosporium* being the most popular [99]. Inflammatory diseases such as Crohn’s disease and ulcerative colitis are linked to mycobiome dysbiosis [100] However, no link was reported between the role of mycobiome and PCOS by Illiev et al. [101]. Mihms and coworkers, showed that the genera *Thermomyces* and *Saccharomyces* have links with host metabolism. Analysis of gut mycobiome by using random forest machine learning models and performing variable importance analysis to identify critical fungal taxa, revealed that a positive correlation existed between *Thermomyces* and weight gain, *Cladospoirum* and serum triglyceride concentration, and *Saccharomyces*, as well as *Aspergillus*, with fasting ghrelin levels. Moreover, the production of secondary bile acids was previously attributed to various microbionts that were considered essential for producing metabolic hormones such as leptin, resistin, ghrelin, and Glucagon like peptide-1 (GLP-1). However, it is evident now that fungi such as *Fusarium*, *Aspergillus*, and *Penicillium* also produce secondary bile acids [101]. A study pointed out that *Candida*, *Nakaseomyces*, and *Penicillium* are the most abundant genera in obese patients. The relative abundance of strains of the phylum *Zygomycota* and the class *Eurotiomycetes*, family *Mucoraceae*, was negatively related to serum total cholesterol, LDL-cholesterol, and fasting triglycerides. The relative abundance of strains from the *Dipodascaceae* family, on the other hand, was positively linked to serum total cholesterol and fasting triglycerides. *Eupenicillium* was negatively associated with homeostatic model assessment (HOMA) value [102]. Of all fungal species, *Candida* has successfully been identified from the intestine of healthy people [103]. Some research has also revealed a relationship between the abundance of *Candida spp*. with diabetes and inflammation in the gastrointestinal tract [104,105]. However, more extensive studies are required to further elucidate the role of mycobiome in PCOS. 

### 3.3. Virus Involved in PCOS

Intestinal bacteriophages have been identified as the gut virome’s primary component, accounting for over 90% of its makeup [106]. A prospective virome investigation in a group of 19 children was conducted before and after the onset of islet autoimmunity. There were no significant alterations in the gut virome before or after the formation of islet autoimmunity or after the onset of diabetes [107]. The virome may or may not have a role in the pathogenesis of PCOS or any other metabolic disease and its role remains speculative. However, manipulation of virome has been found to benefit for managing intestinal inflammation [108]. Toll like receptor-3,7 (TLR3) and (TLR7) recognized resident viruses promote intestinal homeostasis by secreting anti-inflammatory cytokines such as interferon-beta (IFN-β) produced mainly by plasmacytoid dendritic cells [109]. The abundance of pathogenic and opportunistic viruses in the guts of PCOS patients is still unreported. In the absence of protective immunity, a significant number of viral species are likely to coexist and maybe transiently removed just to resurface later. Prospective longitudinal studies aimed at characterizing the dynamics of the gut virome at a steady state in healthy and PCOS are needed to understand better the driving forces that shape the gut microbiome, to monitor and ideally predict pathogenesis associated with these viruses. 

## 4. Mechanistic Pathways in PCOS

### 4.1. Correlation between Gut Microbiota and Hyperandrogenism

While the cause of PCOS is not clear in absolute terms, two different but related theories have been proposed to link the development of hyperandrogenic PCOS traits to changes in the gut flora [110]. One study pointed out that gut dysbiosis caused by a high-fat diet and a high carbohydrate diet influences the gut barrier function, leading to insulin resistance, hyperandrogenism, and dysfunction of the ovaries. This hypothesis strongly points towards the diet and gut dysbiosis as driving forces from where pathogenic features of PCOS related to hyperandrogenism emerge [111]. This hypothesis, however, does not consider that despite differences in diet also, occurrence of PCOS is observed [2]. Qi and his colleague transplanted fecal microbiota of PCOS women into antibiotic-treated mice and noticed reproductive and metabolic abnormalities in the recipients. In addition to hyperandrogenism, the PCOS control mice experienced altered estrous cyclicity and decreased ovulation, as evidenced by a reduction in the number of corpora lutea in the ovaries and ovarian cysts’ formation along with a reduction in fertility [112]. This study reveals that dysbiotic gut microbiota from women with PCOS is sufficient to elicit a PCOS-like phenotype in mice, supporting the theory that changes in the gut microbiome may play a causative role in this condition. Another hypothesis is based on the fact that testosterone could affect the gut microflora through a direct effect as a substrate for gut microbial enzymes and an indirect influence via activation of host androgen receptors or immune system regulation [113,114]. Since it is not pragmatic to induce gut dysbiosis to cause hyperandrogenism in a clinical setting, several studies have deciphered this link between hyperandrogenism and gut microflora in animal models. In two studies, the genus *Collinsella* was found to be positively linked with testosterone levels in women [78,115]. The genus *Bacteroides* was also positively correlated with testosterone levels [88]. The genus *Prevotella*, on the other hand, was found to be negatively correlated with testosterone in another study [116].

Further, hyperandrogenism may potentially cause dysbiosis of the gut microbiome, according to evidence from animal models. Letrozole, a nonsteroidal inhibitor of aromatase and used as an inducing agent for PCOS, has been used to determine any changes in gut microbiota. Treatment with letrozole was found to result in a change in the abundance of *Bacteroidetes* and *Firmicutes*. The result was in line with the fact that hyperandrogenemia in PCOS alters the gut microbiota independent of diet [44]. Another study pointed out that letrozole treatment resulted in greater relative abundances of *Coprococcus*, *Allobaculum*, *Bifidobacterium*, and *Ruminococcaceae* genus members in pubertal female mice [117]. Surprisingly, these letrozole-induced changes in the gut microbiome appear to be activational rather than organizational, as stopping the letrozole treatment after induction of gut dysbiosis was found to restore gut bacterial diversity.

Another study pointed out that the number of *Bacteroidetes* increased after administration of dihydrotestosterone (DHT) (inducing agent for PCOS) and resulted in a lower abundance of *Proteobacteria, Spirochaetae*, and *Verrucomicrobia*. All these results suggest the strong relationship of testosterone with the gut microbiota. However, it is still speculative whether testosterone exerts its effects directly or indirectly in the androgenic tissue. Further, investigations on inhibition of androgen receptors using antiandrogens such as spironolactone, cyproterone and androgen receptor knockdown in specific host tissues will determine which androgen action sites are essential for gut dysbiosis in women. 

### 4.2. Correlation between Gut Microbiota and Energy Absorption

Intestinal fermentation of complex carbohydrates requires interactions among members of the microbiome community, which comprise both nutritionally specialized and generally adapted species. Many genes, encoding carbohydrate-active enzymes, are found in certain dominating species, particularly among the *Bacteroidetes*. This enables them to quickly switch between different energy sources in the gut, based on the source availability. This is attributed to their sophisticated sensing and regulatory processes that control gene expression [118].. Kocelak and coworkers studied the resting energy expenditure (REE), body composition, and the gut microbial population in 50 obese and 30 lean, healthy, weight-stable subjects. They found that obese people had a significantly higher total microbial count, but no significant variations in the *Bacteroidetes*/*Firmicutes* ratio was found, which had previously been reported to be lower in obese people in other investigations [119]. The gut microbiota can aid in the usage of calories from consumed foods. Turnbaugh et al. found that the obese (ob/ob) mouse microbiota, whether present in ob/ob or gnotobiotic mice, had a higher potential to harvest energy from the food. More importantly, transplanting an obese mouse’s caecal microbiota into gnotobiotic mice resulted in enhanced energy harvesting, as well as increased intestinal monosaccharide absorption. As a result, hepatic lipogenesis got enhanced, and hepatic lipoprotein lipase (LPL) as well as sterol regulatory element-binding proteins (SREBPs) were stimulated [120]. The link between gut microbiota, energy metabolism, and obesity was validated in this investigation. Over half of PCOS individuals were overweight or obese, according to data in the study [121].

Furthermore, both gut microbiota richness and phylogenetic diversity in PCOS patients are dramatically reduced [122]. A study compared the gut microbiota composition of groups of obese women with PCOS, non-obese women with PCOS, obese women without PCOS, and normal controls to further evaluate the association between obesity, PCOS, and gut microbiota. The findings indicated that obese and non-obese women with PCOS had significantly different levels of β diversity [122]. It was hypothesized that gut microbiota might uniquely influence weight and energy metabolism in patients with PCOS. From the initial studies, it can be assumed that gut microbiota disturbances can hasten the progression of PCOS by impairing energy absorption also.

### 4.3. Correlation between Gut Microbiota and Short Chain Fatty Acid Metabolism

Short-chain fatty acids (SCFAs), the primary metabolites generated in the colon due to bacterial fermentation of dietary fibers and resistant starch, are thought to play a critical role in neuro-immunoendocrine control. However, the fundamental processes through which SCFAs may affect PCOS physiology, hitherto remains unknown [123]. Den Besten et al. reported that SCFAs activate the peroxisome proliferator-activated receptor gamma (PPAR-γ) in the liver and muscles, controlling glucose absorption and fatty acid oxidation. Additionally, gut microbiota may alter insulin sensitivity through branched-chain amino acid-mediated inflammatory responses (BCAAs) [124,125]. Another study revealed that the more BCAAs the individuals consumed, the greater is their risk of developing type 2 diabetes [126]. The reason for this was thought to be 3-hydroxybutyrate (3-HIB), a valine catabolite that increases fatty acid absorption in the muscle tissue. This, in turn, results in fat buildup and insulin resistance [127]. Another study linked gut microbiota and BCAA metabolism. *Prevotella* was discovered to be involved in BCAA production. Mice that were given a high-fat diet had increased BCAA levels in their blood after two weeks, while, after three weeks, they developed insulin resistance. This animal model accurately mimiced the metabolic condition of PCOS patients who are obese or eat a high-fat diet. SCFAs may decrease the production of the appetite-stimulating hormone (ASH) by the stomach mucosa when combined with free fatty acid receptor-2, and 3 (FFAR-2/3). These are generally thought to be the contributing factors [128].

ASH not only suppresses gonadotropin hormone-releasing hormone (GnRH) secretion but also sex hormone transformation [129]. Moreover, ASH is known to inhibit the production of aromatase CYP19A1 in adipo-stromal cells, preventing androgen conversion to estrogen. Reduced ASH means more androgen levels [130]. A meta-analysis reported that PCOS patients had lower ASH levels than non-PCOS women [131]. An altered SCFA metabolism induced by an abnormal gut microbiota seems to be linked to IR and hyperandrogenemia in PCOS. To make matters worse, high testosterone levels may exacerbate the problem leading to a vicious cycle.

### 4.4. Correlation between Gut Microbiota and Lipopolysaccharide (LPS) Metabolism

Certain Proteobacterial lipopolysaccharides (pro-inflammatory LPS, P-LPS) have been reported to cause septic shock or even death in animals as well as people. Although P-LPS from various bacterial species have comparable structural and chemical characteristics, they have modest and variable immune activation actions. However, a subset of LPS molecules generated by some microbes, such as *Bacteroidetes*, have a muted or even antagonistic effect in initiating pro-inflammatory reactions affecting insulin sensitivity and leading to insulin resistance [132]. In one of such studies, mice received either a low-fat or high-fat diet, and the change in gut microbiota was noticed. After four weeks, mice consuming a high-fat diet were found to be obese and exhibited symptoms of insulin resistance. Their blood LPS concentration was 2–3 times greater than that of the control group and their gut microbiota composition was also reported to be altered [133].

The LPS pathway is also known to promote insulin resistance and obesity in PCOS when gut microbiota is aberrant [134]. Low pro-inflammatory factor release over time, may cause hyperandrogenism and obesity in PCOS [135]. Elevated androgen levels increase inflammation in PCOS patients. A study demonstrated that testosterone increases the response of 3 T3-L1 adipocytes to LPS, resulting in enhanced production of IL-6, which validates this hypothesis [136]. Gut microbiota mediates insulin resistance through LPS. Some experts claim insulin resistance is the key to PCOS’s aberrant metabolism and creates a persistent inflammatory state.

### 4.5. Correlation between Gut Microbiota and Choline Pathway

Higher plasma levels of trimethylamine N-oxide levels have been reported in PCOS without the association of hyperandrogenism [137]. PCOS is widely reported to be associated with a high cardiovascular risk [138]. The bacterial formation of trimethylamine (TMA) has been linked to cardiovascular disease. Until recently, the metabolic mechanisms involved in bacterial TMA production were unclear [139]. The choline-TMA lyase, CutC protein, needs a glycyl radical to break the carbon-nitrogen bond, while the carnitine-TMA lyase, CntA protein, uses a mononuclear iron in the active center. CutC is part of a broader enzyme family that includes glycerol dehydratase and pyruvate formate lyase. CntA is a novel Rieske-containing oxygenase with substantial sequence similarity to numerous Rieske protein family members. While these enzymes responsible for TMA production have been identified, finding the microbiota species responsible for TMA generation in the human gut still remains a challenge and even more so, for those involved in PCOS. However, there are some studies that highlight its role in PCOS [140].

First, TMA is metabolized to trimethylamine oxide (TMAO) which regulates cholesterol metabolism, IR, platelet aggregation, and inflammation [141]. Further, a rise in TMAO levels due to increased TMA-producing bacteria in the gut, may cause atherosclerotic lesions and cardiovascular disease. PCOS patients often have hyperlipidemia and IR which may cause endothelial damage and thrombosis. These metabolic anomalies raise coronary heart disease risk. Nevertheless, the chemical mechanism causing this diseased condition is unknown [142]. The link between gut microbiota and its impact on cardiovascular health through the choline route is reported, but it is yet unknown whether this mechanism contributes to cardiovascular problems in PCOS patients.

### 4.6. Correlation between Gut Microbiota and Bile Acid Pathway

The gut microbiota influences bile acid composition and metabolism in the liver through Farnesoid X receptor and G protein-coupled membrane receptor 5 signaling [143]. The gut microbiota acts by decreasing the activity of cholesterol 7-hydroxylase (CYP7A1), which controls fat synthesis and triglyceride TG levels [144]. Abnormal lipid metabolism is reported in women with PCOS, with a prevalence of up to 50%, and it mostly appears as elevated levels of triglycerides (TG), total cholesterol (TC), low-density lipoprotein (LDL), and hormone-sensitive lipase (HSL) [145]. On the other hand, Qi and coworkers observed that PCOS patients had substantially lower glycodeoxycholic acid (GDCA) and tauroursodeoxycholic acid (TUDCA) levels than the positive control group. *Bacteroides forsythus* correlated adversely with GDCA and TUDCA and uncoupled the binding bile acids produced in the liver. These findings indicate that *B. forsythus* is present in the gut microbiota of PCOS patients, influencing GDCA and TUDCA metabolism [112]. Deoxycholic acid also improves the PCOS phenotype by inducing IL-22 production by intestinal lymphocytes through GATA binding protein 3. This is in line with decreased IL-22 levels in PCOS individuals [146]. Gut microbiota may also alter insulin sensitivity through an inflammatory response mediated by bile acids [147]. Based on previous research, we conclude that a negative alteration in gut microbiota alters bile acid metabolism and chronic inflammation in PCOS patients.

### 4.7. Correlation between Gut Microbiota and Intestinal Permeability

The mechanical component of the intestinal barrier
is the mucosal lining. Tight junctions (TJs) control paracellular or
transcellular bacterial translocation. Increased permeability indicates barrier
function damage, as seen by dual sugar absorption studies. Obesity has been
linked to increased intestinal permeability [148].
This, in turn, corresponds with insulin resistance (HOMA Index) [149] and is exacerbated by liver damage [150]. TNF and INF have been found to decrease the
expression of tight junction proteins ZO-1 and occluding, resulting in tight
junction breakdown and increased intestinal permeability. The occluding and
tricellulin loss also alters the tight junction proteins claudin and zonula
occludens [151]. PCOS is classified as a
chronic inflammatory disorder, and the persistent low release of
pro-inflammatory factors results in ongoing damage, leading to increase in
intestinal permeability. This pathological process, in turn, is intimately
linked to the gut microbiota. It has been suggested that one of the pathogenic
mechanisms of PCOS could be that obesity and a high-sugar, high-fat diet with
low dietary fiber promotes gut microbiota imbalance, destroying the junction
between intestinal epithelial cells and the reduction in the expression of ZO-1
and occludin [152]. Besides, the expression of
cannabinoid 1 (CD-1) is suppressed, which has the potential to act on tight
junction proteins, resulting in the development of “leaky gut.” By this
process, LPS from gram-negative bacteria enters the body circulation,
contributing to an antigen-antibody reaction that activates the immune system,
induces chronic inflammation, impairs the function of insulin receptors, and
thereby increases the level of insulin, resulting in elevated androgen levels
and irregular follicular development leading to PCOS [153].

### 4.8. Correlation between Gut-Brain Axis

Enteric microbiota seems to have a role in the gut-brain axis, interacting with intestinal cells, the enteric nervous system (ENS), and the central nervous system (CNS) through neuroendocrine and metabolic pathways. Dysbiosis occurs in functional gastrointestinal disorders related to mood disorders and gut-brain axis dysfunction [154]. GLP-1, for example, is disordered in PCOS patients [155]. GLP-1 affects the gastrointestinal system and the CNS via the vagus nerve; GLP-1 plays a critical role in multiple functions such as stalling gastric emptying time, lessening appetite, increasing satiety, promoting pancreatic islet cell proliferation, and stimulating insulin [156]. PCOS patients often suffer from depression, social phobias, anxiety, and aggressiveness. These symptoms are linked to a faulty brain-gut axis. Enterochromaffin cells, a type of intestinal endocrine cells, produce most of the body’s 5-HT. This impacts brain growth, stress response, and vigorous activity such as anxiety and sadness [157]. Compared to normal mice, PCOS mice were reported to exhibit decreased 5-HT, norepinephrine, and dopamine [158]. The intestinal microbiome has been shown to disrupt host immune regulation through the brain-gut axis. The structural foundation for immune control is in intestinal lymphoid tissues, which comprise 70–80% of the body’s immune cells [159]. Using peripheral blood samples from individuals with PCOS-related infertility, Lang and co-workers discovered that an imbalance of Th1 and Th2 cells leads to poor oocyte quality and ovulation problems with poorer pregnancy rates [160].

Further, researchers discovered a link between gut microbiota imbalance and the Th1/Th2 ratio, suggesting that gut microbiota and helper T cell balance are interdependent and mutually limiting. There is no experimental proof that gut microbiota controls immunity through the brain-gut axis. Therefore, further studies are needed to reach any definite conclusion in this regard [161]. A figure explaining the role of the gut microbiota and its pathways is shown in Figure 1.

## 5. Potential Treatment Strategies for PCOS

The current PCOS therapy focuses on amelioration of the patient’s symptoms. These symptoms include menstruation problems and infertility. Clomiphene is the first-line therapy for infertile women [162]. Recent clinical trials show that letrozole (in the right dose) improves ovulation rate, monofollicular formation, mean endometrial thickness, and pregnancy rate better than clomifene citrate [163]. Metformin is another drug used in the treatment of PCOS and is also used to help women conceive. Metformin lowers insulin and testosterone levels, affecting the ovulatory cycle and periods. It also raises sex hormone-binding globulin (SHBG) and improves lipid profiles [164]. Other infertility treatments include gonadotropin-stimulating ovulation or laparoscopic surgery [165]. The other pharmacological therapy for PCOS is the usage of oral contraceptives such as drospirenone for decreasing hirsutism and reducing testosterone levels [166]. There is no current treatment strategy for PCOS that regulates gut microbiota. We have outlined possible gut microbiota therapy approaches for PCOS along with the new clinical treatment suggestions.

### 5.1. Fecal Microbiota Transplant (FMT)

Fecal microbiota transpant is the transportation of gut microbiota from healthy donors’ feces into patients’ small intestines through oral or rectal routes [167]. In light of the hypothesis that gut dysbiosis contributes to PCOS symptoms, treatment with an FMT derived from healthy donors or representative microbes from a healthy gut is expected to be a helpful to re-diversify the gut microbiome [9]. Performing an FMT from healthy rats into a letrozole-induced PCOS rat model led to a reduction in androgen levels, better estrous cycles, improved ovarian shape, higher levels of *Lactobacillus*, and *Clostridium* species, and a decline in *Prevotella* species, according to one study conducted [50]. Co-housing with healthy, placebo-treated mice reduced PCOS reproductive and metabolic symptoms in letrozole-treated mice and altered the relative abundance of *Coprobacillus* and *Lactobacillus* in another study [168]. Transplantation of fecal microbiota of PCOS women into antibiotic-treated mice led to reproductive and metabolic abnormalities in the recipient mice. They exhibited altered estrous cyclicity, decreased ovulation and a reduction in fertility [112]. FMT, therefore, may be a potential treatment option for PCOS, but further research is needed.

### 5.2. Prebiotics

In the gut, microorganisms break down prebiotics. The breakdown products of prebiotics include short-chain fatty acids that have asignificant role in control of inflammation [169]. Prebiotics having positive effects on human health include fructooligosaccharides and galactooligosaccharides [170]. Prebiotics have been shown to enhance microbial fermentation, decrease appetite, and lower post-meal plasma glucose intake. Research has looked at the impact of prebiotics on various metabolic disorders, including diabetes and obesity [171]. Some investigations have shown that prebiotics may enhance microbial fermentation while decreasing hunger and plasma glucose absorption after meals. A study showed that intake of prebiotics increased the abundance of *Bifidobacteria* in the colon and enhanced the production of GLP-1 by colon L-cells, resulting in improved insulin resistance [172]. Based on these results, researchers are looking into the therapeutic effects of prebiotics on PCOS. Another study evaluated the role of resistant dextrin on PCOS. For three months, women with PCOS and women without the disease were given resistant dextrin, a glucose polysaccharide digested in the colon by bacteria rather than absorbed in the small intestine. Free testosterone levels, hirsutism, the time between menstrual cycles, fasting blood glucose, and lipid profile were all reduced by resistant dextrin [173]. In dehydroepiandrosterone (DHEA) treated mice, who were fed a high-fat diet, the prebiotic inulin was demonstrated to ameliorate intestinal dysbiosis, reduce testosterone, and boost estrogen levels while improving ovarian morphology and weight gain [174]. According to the gut microbiota sequencing performed, *Bifidobacteria* were found to increase in the inulin group compared to the standard group, but *Proteobacteria* and *Helicobacter* levels declined. Prebiotics, thus, play a significant role in the management of PCOS by regulating the gut microbiota. However, more research leading to an understanding of the type, amount and duration of administration of fermentable dietary fiber to achieve benefits in PCOS is needed.

### 5.3. Probiotics

Probiotics are defined by the FAO and WHO (Food and Agriculture Organization of the United Nations and World Health Organization) as “live microorganisms that bestow a health benefit on the host when provided in suitable levels” [175]. *Lactobacillus*, in particular, is a probiotic that plays a crucial role in immunomodulation in the intestinal mucosa [72]. In clinical and experimental research, *Lactobacillus* strains have been shown to effectively prevent and treat antibiotic-associated diarrhea, traveler’s diarrhea, and illnesses caused by intestinal pathogens. *Lactobacillus* administered to rats given letrozole treatment, lowered testosterone levels, enhanced estrous cyclicity, normalized ovarian shape, increased *Lactobacillus* and *Clostridium* species, and decreased *Prevotella* species [50]. The effects of probiotic combinations on PCOS characteristics have been reported in several studies. In a DHT-induced PCOS rat model, a combination of *Bifidobacterium*, *L. acidophilus*, and *E.faecalis* enhanced reproductive and metabolic functioning, as well as gut microbiome alpha diversity [176]. Another study indicated that supplementation with *L. acidophilus*, *L. casei*, and *B. bifidum* (2 × 10^9^ CFU/g each) significantly increased serum levels of SHBG, reduced total serum testosterone levels, reduced modified Ferriman–Gallwey (mFG) scores, and improved chronic inflammatory states as indicated by decreased level of IL-6 [177].

Current research on probiotic supplementation has certain limitations. Probiotics employed in each study varied greatly and there were no standard dosages utilized. These issues need to be addressed in future studies. The use of various probiotics and their metabolic outcomes related to PCOS is highlighted in Table 3.

### 5.4. Polyphenols

Recent research suggests that dietary phenolic compounds reaching gut bacteria and the aromatic metabolites produced by them can alter and change the microflora community by acting as prebiotics and antimicrobials against pathogenic intestinal microflora alleviating the PCOS disease state [186]. Polyphenols may be transformed into bioactive chemicals by colonic bacteria, altering gut ecology and human health. In animal and human studies, prescribing quantities of a specific polyphenolic chemical have been demonstrated to alter the gut microflora composition, inhibiting certain bacterial groups while allowing some others to thrive in the ecosystem’s available niche [187]. Several studies investigating the effects of polyphenols in treating PCOS in animal models and patients have been conducted, with promising results. According to one study, resveratrol therapy reduced reactive oxygen species (ROS) generation and increased insulin sensitivity in ovarian tissue in rats induced with testosterone enanthate-induced PCOS [188]. According to another study, quercetin successfully restored PCOS-induced alterations in lipid profile, anti-oxidant status, steroidogenesis, and ovarian architecture in rats with letrozole-induced PCOS [189].

Similarly, curcumin was demonstrated to correct anomalies in glucose and glycosylated haemoglobin levels, lipid profile, serum hormonal profile, and antioxidant activity in letrozole-induced PCOS [190]. Until now, only a few clinical trials have been conducted to assess the efficacy of polyphenols in the treatment of PCOS. Thus, it is indicated that the findings of in vitro and in vivo research warrant further clinical trials with diverse polyphenolic substances. However, there exists very little literature on the role of polyphenols in the modulation of the gut microbiota to treat PCOS. Nevertheless, some common polyphenols and gut microbiota interventions to metabolic parameters exhibited by PCOS that are available in the literature are highlighted in Table 4.

## 6. Conclusions

The gut microbiota is critical in influencing human energy metabolism and is strongly linked to PCOS. Some gut bacteria of genus *Lactobacillus*, *Firmicutes*, and *Bacteroidetes* are linked positively to PCOS development, whereas some species of *Bifidobacterium*, most *Lactobacillus*, and some *Bacteroidetes* display PCOS ameliorating effects. Recently, much focus has been laid on understanding the role of gut microbiota in pathogenesis of PCOS. Alterations in gut microbiota are known to have both positive and negative effects on PCOS development. Metabolites from gut microbiota can promote weight loss through various mechanisms. These include promotion of browning of white adipose tissue, regulation of fatty acid metabolism and hyperandrogenism, decrease in appetite, alleviation of gut inflammation, regulation of lipogenesis genes, decrease in serum levels of triglyceride, cholesterol, and glucose. I It is, however, now important to identify the microorganisms that cause PCOS and those that alleviate its symptoms..

Bacterial consortia and certain stool-derived microbial products that contain fewer taxa of bacteria, viruses, and fungi also seem to be promising approaches [203]. Although no such product is yet in the market, a number of them are in development phases. As these consortia are easier to characterize and standardize, they areanticipated to be better accepted in terms of safety and efficacy.

## Figures and Tables

**Figure 1 pharmaceuticals-16-00197-f001:**
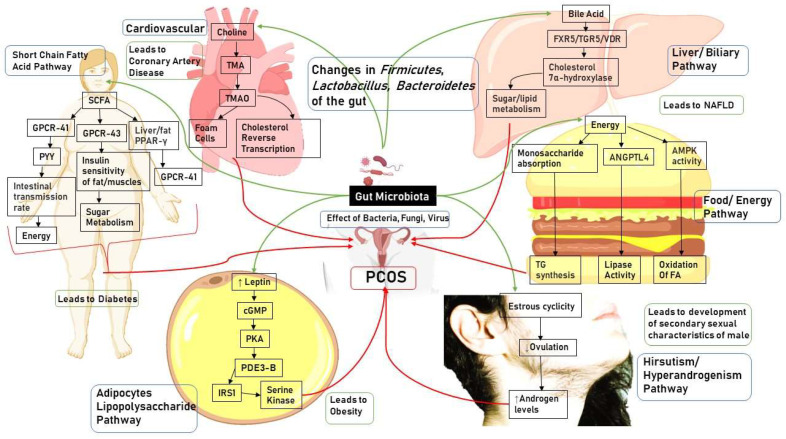
Role of gut microbiota in the cause of PCOS. AMPK: 5′ Adenosine Monophosphateactivated Protein Kinase; ANGPTL4: Angiopoietin-like 4; cGMP: Cyclic Guanosine Monophosphate; FA: Fatty Acids; FXR5: Farnesoid X Receptor 5; GLP-1: Glucagon-Like Peptide-1; GPCR-41/43: G-Protein-Coupled Receptor 41/43; IRS1: Insulin Receptor Substrate 1; LPS: Lipopolysaccharides; NAFLD: Non Alcholic Fatty Liver Disease; PDE3-B: Phosphodiesterase 3B; PKA: PPAR-γ: Peroxisome Proliferator-Activated Receptor γ; PKA: Protein Kinase; PYY: Peptide Tyrosine Tyrosine; SCFA: Short Chain Fatty Acids; TG: Triglycerides; TGR5: G-protein-coupled receptor for bile acids TMA: Trimethylamine; TMAO: Trimethylamine oxide; VDR: Vitamin D receptor; ZO-1: Zonula Occludens-1.

**Table 1 pharmaceuticals-16-00197-t001:** Recent studies on gut microbiota and endocrine function.

Hormone	Model	Finding	Microbiota studied	Reference
Insulin	Honey Bees	Changes in insulin signaling reduces gut pH and gustatory response	*Gilliamella apicola* and *Lactobacillus* spp.	[24]
Testosterone, estradiol	Human	Correlation of diversity of microbiota with sex hormones	*Acinetobacter*, *Dorea*, *Ruminococcus*, and *Megamonas*, *Bacteroidetes* and *Firmicutes*	[22]
Adrenocorticotropic	Wistar rats	Role of gut microbiota in depression	*Desulfovibrionales*, *Desulfovibrio*, *Klebsiella*, *Burkholderiales*, and *Bifidobacterium*	[25]
Ghrelin	Wistar rats	Short-chain acid-producing bacteria and its effects on gut microbiota	*Turicibacter*, *Brevibacterium*, *Parasutterella*, and *Oscillibacter*	[26]
Juvenile hormone III	*Riptortus pedestris*	A more significant number of eggs produced as a result of Burkholderia gut symbiont	*Burkholderia*	[27]
Adrenocorticotrophic hormone	Wistar rats	Changes in the microbiome diversity	*Akkermansia* and *Lactobacillus*	[28]
Epinephrine and Norepinephrine	Bacterial growth	Stress hormones can affect the growth of anaerobic bacteria in the gut	*Fusobacterium nucleatum*, *Prevotella*, *Porhyromonas* spp., *Tanerella forsythia*, and *Propionibacterium acnes*	[29]
Thyroxine	Mice	Reduced diversity of some bacterial species	Bacteria belonging to families *Lactobacillaceae* and *Bifidobacteriaceae*	[30]
Androgen	Pigs	Influence of host gender on gut microbiota and sex-biased bacterial taxa	Bacteria belonging to families *Veillonellaceae*, *Roseburia*, *Bulleidia*, and *Escherichia*	[31]
Androgen	Wistar rats	Prenatal androgen was associated with the abundance of gut microbiota	*Akkermansia, Bacteroides*, *Lactobacillus*, *Clostridium*, *Nocardiaceae*, and *Clostridiaceae*	[32]
Glucagon-like peptide 1	Mice	Intestinal microbiota increases GLP-1 levels	*L. paracasei*, *L. bulgaricus*, and *Streptococcus thermophilus*	[33]
Corticosterone, adrenocorticosterone	Probiotics in rats and humans	Bacteria reduce levels of stress hormones	*L. helveticus* and *B. longum*	[34]

**Table 2 pharmaceuticals-16-00197-t002:** Synbiotic treatment and outcomes related to PCOS.

Synbiotic Treatment	Model	Outcome	Reference
*L. bifidum*, *L. acidophilus*, *L. casei* and inulin	Clinical study	No change in hirsutism	[59]
*Lactobacillus*, *Bifidobacterium*, and Selenium	Clinical study	Improved testosterone and hirsutism	[60]
*L. bifidum*, *L. acidophilus*, *L. casei* and inulin	Clinical study	Improved insulin sensitivity	[59]
*L. acidophilus*, *L. reuteri*, *L. fermentum*, *B. bifidum* and selenium	Clinical study	Increased insulin sensitivity	[61]
*L. plantarum* and inulin	Wistar rats	Decrease hyperglycemia, IR, hyperlipidemia, and ameliorate oxidative stress	[62]
*L. acidophilus*, *L. casei*, *L. rhamnosus* and inulin	Clinical study	Reduced low-density lipoprotein (LDL) and increased high-density lipoprotein (HDL)	[63]

**Table 3 pharmaceuticals-16-00197-t003:** The role of various probiotics and their metabolic outcomes related to PCOS.

Probiotic Used/Dose	Design	Source and Duration	Number of Participants	Outcome	Reference
*L. acidophilus* La5 (4.14 × 106 CFU/g), *Bifidobacterium lactis* Bb12 (3.61 × 10^6^ CFU/g)	Double-blind, placebo	Yogurt, 6 weeks	64	↓ Cholesterol, ↓ LDL-C	[178]
*L. acidophilus* La5 (7.23–1.85 × 10^6^ CFU/g), *Bifidobacterium lactis* Bb12 (6.04–1.79 × 10^6^ CFU/g)	Double-blind placebo	Yogurt, 6 weeks	64	↓ Fasting glucose, ↓serum malondialdehyde concentration, ↑ erythrocyte superoxide dismutase, ↑ glutathione peroxidase	[179]
*L. acidophilus* (2 × 10^9^ CFU) *L. casei* (7 × 109 CFU) *L. rhamnosus* (1.5 × 10^9^ CFU) *L. bulgaricus* (2 × 10^8^ CFU), *B. breve* (2 × 10^10^ CFU), *B. longum* (7 × 10^9^ CFU), *S. thermophilus* (1.5 × 10^9^ CFU)	Single blind, placebo, parallel	Capsule, 8 weeks	54	↑ Serum insulin, ↑ LDL-C, ↑ GSH levels, ↓ serum hs-CRP	[180]
*L. acidophilus*, *L. bulgaricus*, *L. bifidum*, *L. casei*, *L. sporogenes* (1 × 10^8^ CFU)	Single-blind, placebo, parallel	Capsule, 6 weeks	34	↑ HDL-C, ↓ Insulin, ↓ MDA, ↓ IL-6	[181]
*L. sporogenes* (1 × 10^8^ CFU)	Double-blind, placebo, parallel	Bread, 8 weeks	81	↓ Serum insulin levels, ↓ HOMA-IR scores, ↓ HOMA-B, ↑ inflammation markers	[182]
*L. acidophilus*, *B. lactis* (3.7 × 10^6^ CFU/mg)	Double-blind, placebo, parallel	Yogurt, 8 weeks	44	↓LDL-C/HDL-C ratio, ↓ triglycerides	[183]
*L. actobacillus*, *L. helveticus*	Double-blind, placebo control	Yogurt, 12 weeks	41	↓ Blood glucose concentration, ↓serum glucose	[184]
*Lactobacillus* species	Double-blind, placebo control, parallel	Sachet, 12 weeks	136	↓HbA1c, ↓ triglycerides, ↓ insulin resistance	[185]

HbA1c: Hemoglobin A1C; HDL-C: High-density lipoprotein-cholestrol; HOMA: Homeostatic model assessment; hs-CRP: High senstivity c reactive protein; IL: Interleukin; LDL: Low density lipoprotein-cholestrol; MDA- malondialdehyde.

**Table 4 pharmaceuticals-16-00197-t004:** Polyphenol’s role on the gut and metabolic parameters.

Source	Polyphenol	Animal Used, Number of Treatment Controls	Changes in Gut Microbiota	Changes in Metabolic Parameters	Reference
Apple (*Pyrus Malus*)	Procyanidins	C57BL/6J mice, n = 10	↓The ratio of *Firmicutes* to *Bacteroidetes*, ↑ *Akkermansia*, *Bacteroidetes* and *Lactobacillus*	↓Pro-inflammatory factors TNF-α, IL-1β, MCP-1, and chemokine ligand 1, metabolic endotoxemia	[191,192]
Grapes (*Vitis Vinifera*)	Resveratrol	Male Kunming mice, n = 8	↑ *Bacteroidetes* to *Firmicutes* ratio, *Lactobacillus*, and *Bifidobacterium*, ↓ *Enterococcus faecalis*	↓Weight gain and visceral adipose weight	[193]
Berries	Anthocyanins	SD rats, n = 8 Wistar, n = 8 C57BL/6J mice, n = 8	↑ *Akkermansia* and *Desulfovibrio*, *Faecalibacterium*, *Gammaproteobacteria*	↓TNF-α and IL-1β levels,	[194,195,196]
Citrus fruits	Piceatannol	C57BL/6J mice, n = 8	↑ *Prevotella*	↓ Lipid droplets, perilipin 1 protein, and sterol regulatory element-binding protein 1	[197]
Japanese Persimmon (*Diospyros kaki*)	Tannin	SD rats, n = 6	↑ *Bifidobacterium* and *Lactobacillus*, ↓ *Firmicutes*, *Escherichia coli*, and *Enterococcus*	↓Serum lipids and cholesterol	[198]
Turmeric (*Curcuma Longa*)	Curcumin	C57BL/6J mice, n = 6	↑ *Prevotella*, *Bacteroidaceae*, and *Rikenella*	↑Expression of tight junction proteins, gut permeability, ↓NF-κB	[199,200]
Chilli (*Capsicum frutescens*)	Capsaicin	ob/ob mice, n = 5	↑ *Firmicutes* to *Bacteroidetes* Ratio, ↓ *Bacteroides* and *Parabacteroides*	↑Fecal butyrate and plasma total glucagon-like peptide-1 (GLP-1) levels, and ↓total ghrelin, TNF-α, IL-1 β, and IL-6 levels	[201]
Rosemary (*Salvia rosmarinus*)	Carnosic acid	Zucker obese rats, n = 10	↑ *Blautia coccoides* and *Prevotella*	↓ Bodyweight	[202]

C57BL/6J- C57 black 6; MCP-1—Monocyte chemoattractant protein-1; NF-κB—nuclear factor kappa light chain enhancer of activated B cells; ob/ob- obese mice; SD- Sprague Dawley.

## Data Availability

Data sharing not applicable.

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
