# Peer review of "Interplay of Gut Microbiota in Polycystic Ovarian Syndrome: Role of Gut Microbiota, Mechanistic Pathways and Potential Treatment Strategies"

_pharmaceuticals, 2023, doi:10.3390/ph16020197_

Round 1

Reviewer 1 Report

In this manuscript Corrie et al summarize the available evidence regarding the correlations of gut microbiota with PCOS (etiology, symptoms, etc). 

Overall, this is an inclusive review that contains extensive information on the matter in question. However, this is a very long manuscript which is characterized by very poor quality English, making it difficult to read. I believe that extensive editing of English language is needed before this manuscript is suitable for publication. 

Further comments:

- I believe the Introduction section should be altered as follows: the first sentences of paragraph 2 (lines 53-55) can be moved to paragraph 1; paragraph 2 should be brokeen in two new paragraphs, the first presenting data regarding the interactions of the gut microbiome with obesity, inflammation and metabolism, and the second presenting the available data regarding the correlations of the gut microbiome with PCOS. 

- A new section should be added, where the authors briefly present their methodology concerning the literature reseach they conducted in order to collect the necessary references to conduct this review.

- It is mentioned that "Table 1 highlights the recent studies on gut microbiota and its effects on hormonal signaling, related to PCOS". The studies cited in the Table are not all conducted on PCOS/PCOS models. Please rephrase. 

- Lines 118-119: "...a change in the alpha and beta diversity of the bacterial population in the gut is correlated with the occurrence of PCOS". Please define alpha and beta diversity.

- Section 3: I believe this section can be drastically reduced in length, and the authors should only report the findings which are relevant to PCOS.

- Table 3 could be omitted. 

- Paragraph 3.2 ("virus involved in PCOS") should be 3.3.

- I believe that Section 4 ("Gender specific differences in gut microbiota") is not essential and could be removed from the manuscript. 

- In Section 6 ("Treatment Strategies for PCOS"), in the first paragraph, I think that the authors should also mention oral contraceptives as one of the most widely used pharmacologic approaches for PCOS.

- Tables 4 and 5: please include a column with brief information regarding the participants in each study (number of women, PCOS/controls, etc).

Author Response

Comment 1

In this manuscript Corrie et al summarize the available evidence regarding the correlations of gut microbiota with PCOS (etiology, symptoms, etc). 

Overall, this is an inclusive review that contains extensive information on the matter in question. However, this is a very long manuscript which is characterized by very poor quality English, making it difficult to read. I believe that extensive editing of English language is needed before this manuscript is suitable for publication.

Response: We thank the learned reviewer for the sagacious comments, to add value to our manuscript. We`ve tried improving the English. We hope it is now fit for publication.

Comment 2

I believe the Introduction section should be altered as follows: the first sentences of paragraph 2 (lines 53-55) can be moved to paragraph 1; paragraph 2 should be brokeen in two new paragraphs, the first presenting data regarding the interactions of the gut microbiome with obesity, inflammation and metabolism, and the second presenting the available data regarding the correlations of the gut microbiome with PCOS. 

Response: The said changes have been applied.

Comment 3

A new section should be added, where the authors briefly present their methodology concerning the literature reseach they conducted in order to collect the necessary references to conduct this review.

Response: A new section at the end of the introduction has also been added (Line 97-103).

Comment 4

 It is mentioned that "Table 1 highlights the recent studies on gut microbiota and its effects on hormonal signaling, related to PCOS". The studies cited in the Table are not all conducted on PCOS/PCOS models. Please rephrase. 

Response: The said line has been rephrased to “Table 1 highlights the recent studies on gut microbiota and its effects on hormonal signaling”

Comment 5

Lines 118-119: "...a change in the alpha and beta diversity of the bacterial population in the gut is correlated with the occurrence of PCOS". Please define alpha and beta diversity.

Response: It has been added in line 86 and line 87, respectively.

Comment 6

Section 3: I believe this section can be drastically reduced in length, and the authors should only report the findings which are relevant to PCOS.

Response: The said changes have been added wherever necessary.

Comment 7

Table 3 could be omitted. 

Response: Table 3 has been removed and table numbers have been updated accordingly.

Comment 8

Paragraph 3.2 ("virus involved in PCOS") should be 3.3.

Response: The section number has been changed and all trailing section numbers have been updated.

Comment 9

I believe that Section 4 ("Gender specific differences in gut microbiota") is not essential and could be removed from the manuscript. 

Response: The said section has been removed.

Comment 10

In Section 6 ("Treatment Strategies for PCOS"), in the first paragraph, I think that the authors should also mention oral contraceptives as one of the most widely used pharmacologic approaches for PCOS.

Response: We have also mentioned about oral contraceptives in the paragraph (Line 611-612)

Comment 11

Tables 4 and 5: please include a column with brief information regarding the participants in each study (number of women, PCOS/controls, etc).

Response: Two new columns have been added to now Table 4 and 5.

Reviewer 2 Report

This manuscript illustrates in detail the Interplay of gut microbiota in polycystic ovarian syndrome and the Mechanistic Pathways of PCOS that natural sources of polyphenols, probiotics, and fecal microbiota therapy can treat by altering the gut microbiota. The manuscript starts with Gut microbiota and the endocrine system as the topic and then discusses the Role of gut microbiota in PCOS and Gender-specific differences in gut microbiota. The overall manuscript is rigorously written, concise and easy to read, and it is an article with a high academic reference value. Well-suited for publication after partial correction.

1. There have been few citations on polycystic ovarian syndrome and gut microbiota in the past two years. There are only three citations in 2022 and 9 citations in 2021.

2. Formatting issues. Line 73, Line 90, Line 132, Line 222, Line 260, Line 709, Line 615, Line 624(Table 4)

Author Response

Comment 1

This manuscript illustrates in detail the Interplay of gut microbiota in polycystic ovarian syndrome and the Mechanistic Pathways of PCOS that natural sources of polyphenols, probiotics, and fecal microbiota therapy can treat by altering the gut microbiota. The manuscript starts with Gut microbiota and the endocrine system as the topic and then discusses the Role of gut microbiota in PCOS and Gender-specific differences in gut microbiota. The overall manuscript is rigorously written, concise and easy to read, and it is an article with a high academic reference value. Well-suited for publication after partial correction.

Response: We thank the reviewer for his encouraging words. We are elated to know that the reviewer finds it suitable for publication and considers it to be of high academic reference value.

Comment 2

There have been few citations on polycystic ovarian syndrome and gut microbiota in the past two years. There are only three citations in 2022 and 9 citations in 2021.

Response: Please note that, this is a relatively new concept. Therefore, the research, particularly for PCOS and its link to gut microbiota is limited and scarce.

Comment 3

Formatting issues. Line 73, Line 90, Line 132, Line 222, Line 260, Line 709, Line 615, Line 624(Table 4)

Response: We thank the reviewer for his suggestions. The formatting errors have been updated in line 83, line 111-112, line 154, line 250-251, line 292- 293, line 775-776, line 675, line 684

Reviewer 3 Report

In the article submitted by Corrie et al, the authors have conducted a literature review on the influence of the microbiota on polycystic ovarian syndrome (PCOS). In addition, they propose different signaling pathways and mechanisms by which the gut microbiota could improve or worsen PCOS. Finally, they present 4 potential treatment strategies: prebiotics, probiotics, polyphenols, and fecal transplantation.

The article is an extensive review, well written, neat and clear. The graphic abstract is fabulous. The main problem noted by this reviewer is that the authors use numerous articles on PCOS-related pathologies and little literature on the syndrome itself. This means that many of the results presented are quite speculative and will not necessarily occur in PCOS. Moreover, as the authors state on several occasions, more clinical trials are needed to make any statements in this regard.

- For all these reasons, the first advice I propose is the inclusion of the word "potential" in the title of the article: ... "and potential treatment strategies".

- Do the authors consider that the proposed strategies will really serve to treat PCOS or will they be more interesting as adjuvant therapies to existing pharmacological treatments?

- Since many of the stated results have not been demonstrated in randomized clinical trials in women with PCOS, the authors should avoid such categorical sentences as:

Line 40-42: "This review concludes that polyphenols from natural origin, probiotics, and fecal microbiota therapy CAN BE USED to treat PCOS by modifying the gut microbiota". 

Line 375-376: "Obese individuals tend to take more food, which is due to microbial flora in the gut". Obesity has many causes and cannot be stated that way, as it suggests that the only cause is the microbiota.

Line 486-488: "Based on previous research, we conclude that gut microbiota alters bile acid metabolism and chronic inflammation in PCOS patients, causing additional endocrine problems". The gut microbiota does not always cause changes and certainly not always problems.

It is recommended to review the text as a whole to avoid statements such as those in the examples given.

- Reference 24 in Table 1 is not correct. It is indicated that the model is murine and that article studies the composition of the microbiota in women with PCOS (Cross-sectional study). It is necessary to review all the references included in the tables.

Minor comments:

- Line 61: "etc" without commas.

- Line 70: the authors use the abbreviation GI for the first time without previously explaining that it refers to "gastrointestinal".

- Line 73: There is a period before the references.

- Line 93: A period is missing.

- Line 110: Change the period to a comma.

- Table 1: Change the word thyroid (which is not a hormone) to the hormone determined in the study: thyroxine (T4), triiodothyronine (T3)... Similar with tryptophan.

- What is the order of Table 1 based on? All types of studies are mixed. How important do the authors consider some of their references to be if they do not name them in the text?

- Line 180: the authors distinguish between cytokine and IL-6. To which cytokine do they refer when writing it in the singular?

- Line 342: the term "human model" does not seem appropriate to me.

- Line 363-364: "Further, the further investigation..."

- Line 604: Why do the authors cite an article instead of the original Source where FAO and WHO define what a probiotic is? Is the definition given the WHO definition or the updated ISAPP definition?

Author Response

Comment 1

In the article submitted by Corrie et al, the authors have conducted a literature review on the influence of the microbiota on polycystic ovarian syndrome (PCOS). In addition, they propose different signaling pathways and mechanisms by which the gut microbiota could improve or worsen PCOS. Finally, they present 4 potential treatment strategies: prebiotics, probiotics, polyphenols, and fecal transplantation.

The article is an extensive review, well written, neat and clear. The graphic abstract is fabulous. The main problem noted by this reviewer is that the authors use numerous articles on PCOS-related pathologies and little literature on the syndrome itself. This means that many of the results presented are quite speculative and will not necessarily occur in PCOS. Moreover, as the authors state on several occasions, more clinical trials are needed to make any statements in this regard.

Response: We thank the learned reviewer for his kind and sagacious comments.

Comment 2

For all these reasons, the first advice I propose is the inclusion of the word "potential" in the title of the article: ... "and potential treatment strategies".

Response: The word potential has been updated in the title of the manuscript.

Comment 3

Do the authors consider that the proposed strategies will really serve to treat PCOS or will they be more interesting as adjuvant therapies to existing pharmacological treatments?

Response: The proposed strategies could be used to treat PCOS. Extensive research is still warranted. However, there is no harm in using them as adjuvant therapies as well.

Comment 4

Since many of the stated results have not been demonstrated in randomized clinical trials in women with PCOS, the authors should avoid such categorical sentences as:

Line 40-42: "This review concludes that polyphenols from natural origin, probiotics, and fecal microbiota therapy CAN BE USED to treat PCOS by modifying the gut microbiota". 

Line 375-376: "Obese individuals tend to take more food, which is due to microbial flora in the gut". Obesity has many causes and cannot be stated that way, as it suggests that the only cause is the microbiota.

Line 486-488: "Based on previous research, we conclude that gut microbiota alters bile acid metabolism and chronic inflammation in PCOS patients, causing additional endocrine problems". The gut microbiota does not always cause changes and certainly not always problems.

It is recommended to review the text as a whole to avoid statements such as those in the examples given.

Response: We thank the reviewer for bringing this to our attention, such statements have been removed or rephrased accordingly.

Comment 5

Reference 24 in Table 1 is not correct. It is indicated that the model is murine and that article studies the composition of the microbiota in women with PCOS (Cross-sectional study). It is necessary to review all the references included in the tables.

Response: We thank the reviewer for bringing this to our attention. It has been updated accordingly.

Minor comments:

Comment 6 

Line 61: "etc" without commas.

Response: Updated in line 69

Comment 7

 Line 70: the authors use the abbreviation GI for the first time without previously explaining that it refers to "gastrointestinal".

Response: Updated in Line 80

Comment 8

Line 73: There is a period before the references.

Response: Updated in line 83

Comment 9

Line 93: A period is missing.

 Response: Updated in line 114

Comment 10

Line 110: Change the period to a comma.

Response:

Comment 11

Table 1: Change the word thyroid (which is not a hormone) to the hormone determined in the study: thyroxine (T4), triiodothyronine (T3) Similar with tryptophan.

Response: Since the tryptophan section was confusing, it has been removed.

Comment 12

What is the order of Table 1 based on? All types of studies are mixed. How important do the authors consider some of their references to be if they do not name them in the text?

Response: Table 1 highlights the studies of gut microbiota in hormone signalling carried out in various models. We consider it to be important.

Comment 13

Line 180: the authors distinguish between cytokine and IL-6. To which cytokine do they refer when writing it in the singular?

 Response: The changes have been made in line 211-212

Comment 14

Line 342: the term "human model" does not seem appropriate to me.

Response: The change has been made in line 386-387

Comment 15

Line 363-364: "Further, the further investigation..."

Response: The change has been made in line 410

Comment 16

Line 604: Why do the authors cite an article instead of the original Source where FAO and WHO define what a probiotic is? Is the definition given the WHO definition or the updated ISAPP definition?

Response: The updated reference has been added. (Reference 196)

Round 2

Reviewer 1 Report

The authors have adressed all my comments, and I believe that the manuscript has been improved substantially. 

Please note that spironolactone is not an oral contraceptive and, therefore, the sentence: "oral contraceptives such as Spironolactone" (line 605) should be rephrased.

Author Response

Comment 1:

The authors have addressed all my comments, and I believe that the manuscript has been improved substantially. 

Response: Thanks for accepting our revision.

Comment 2:

Please note that spironolactone is not an oral contraceptive and, therefore, the sentence: "oral contraceptives such as Spironolactone" (line 605) should be rephrased.

Response: Corrected as suggested.